# Predictive Value of Cortical Thickness Measured by Ultrasonography for Renal Impairment: A Longitudinal Study in Chronic Kidney Disease

**DOI:** 10.3390/jcm7120527

**Published:** 2018-12-07

**Authors:** Shotaro Hoi, Tomoaki Takata, Takaaki Sugihara, Ayami Ida, Masaya Ogawa, Yukari Mae, Satoko Fukuda, Chishio Munemura, Hajime Isomoto

**Affiliations:** Division of Medicine and Clinical Science, Faculty of Medicine, Tottori University, Yonago, Tottori 683-8504, Japan; wrrxt597@yahoo.co.jp (S.H.); sugitaka2002@gmail.com (T.S.); idaayami1991914@yahoo.co.jp (A.I.); dm.ogawa29@gmail.com (M.O.); yuuchanfront@gmail.com (Y.M.); maetasa@med.tottori-u.ac.jp (S.F.); chishiom@med.tottori-u.ac.jp (C.M.); isomoto@med.tottori-u.ac.jp (H.I.)

**Keywords:** ultrasonography, kidney size, cortex, CKD risk factors, kidney function

## Abstract

Background: Kidney size is associated with renal function, however it is not elucidated whether kidney size is a risk for the progression of chronic kidney disease. The aim of this study was to investigate the predictive value of morphological evaluation of kidney size by ultrasonography for the progression of renal dysfunction. Methods: Morphological parameters including kidney length, volume, cortical thickness, and medullary thickness were measured by ultrasonography in 87 patients with chronic kidney disease, and adjusted by body size. Renal functions at baseline and after 2 years were measured and the associations of morphological parameters to decline in renal function over 2 years were analyzed. Results: Height-adjusted cortical thickness was correlated to decline in renal function (*r* = 0.426, *p* < 0.001). Height-adjusted cortical thickness could predict renal dysfunction with the area under the curve of 0.786, and height-adjusted cortical thickness of 4.0 mm/cm was a cut off value with a sensitivity of 72.5% and a specificity of 80.0% for the risk of a more than 30% decline in renal function or initiation of dialysis. Conclusions: We provide new insights into the utility of measuring cortical thickness by ultrasonography for predict future renal impairment.

## 1. Introduction

Chronic kidney disease (CKD), defined by reduced glomerular filtration rate (GFR), is a global health burden. CKD is a risk factor for all-cause mortality, cardiovascular events, and end-stage kidney disease (ESKD) [1,2]. Regardless of the etiology, renal function in patients with CKD progressively decline, and the risk of cardiovascular or cerebrovascular events increase with the stage of CKD [3]. Creatinine or inulin is used for the assessment of GFR, however it is difficult to assess renal parenchymal damage before the decline in GFR because of the compensation by residual glomeruli. Therefore, it is necessary to establish simple indices for detecting the initial stages of decline in renal function and predicting the risk for renal dysfunction.

Ultrasonography is a noninvasive and convenient modality widely used for managing kidney disease. As kidney atrophy occurs with the progression of CKD, measurement of kidney length using ultrasonography is routinely performed. Some researchers further investigated the association of kidney function and morphology, and showed that kidney parenchymal thickness measured by ultrasonography was correlated to kidney function [4]. Kidney parenchyma is composed of cortex and medulla [5]. Glomeruli, that define GFR, are distributed in the cortex. Therefore, evaluation of kidney cortex by ultrasonography would be useful. Actually, strong correlation of kidney function to cortical thickness measured by ultrasonography have been reported [6], and findings such as thinning of kidney cortex and obscurity of corticomedullary junction are peculiar to ESKD.

The importance of detailed morphological assessment of kidney size is being recognized, and a cross-sectional study using enhanced computed tomography demonstrated an age-related morphological change in kidney cortex and medulla [7]. However, it is not fully elucidated whether kidney size is a risk factor for CKD and whether morphological assessment has a predictive value for progression of renal dysfunction. It would contribute to better understanding and improve clinical practice of managing CKD to clarify the usefulness of morphological assessment to predict progression of kidney dysfunction. The purpose of this study was to investigate the prognostic value of morphological evaluation of kidney size by ultrasonography on the progression of renal dysfunction.

## 2. Materials and Methods

### 2.1. Study Population

This study included 137 patients who underwent abdominal ultrasonography between April 2014 and March 2015. Among them, patients with diabetes mellitus, unilateral kidney, renal tumor, hydronephrosis, solitary cyst greater than 4 cm in diameter, polycystic kidney disease, acute kidney injury defined by risk of renal dysfunction, injury to the kidney, failure of kidney function, loss of kidney function and end-stage renal disease (RIFLE) criteria [8], naive active glomerulonephritis indicated by persisting hematuria and/or proteinuria, proteinuria more than 0.5 g/day [9], nephrotic syndrome, transplanted kidney and on maintenance dialysis were excluded from this study. Since a significant difference in kidney size between right and left kidneys indicate the laterality of renal function, patients with asymmetry in length more than 2 cm were also excluded from the study according to a previous report [10] (Figure 1). The height and body weight of all patients were measured, and body surface area (BSA) was calculated as previously described [11]. Serum creatinine of the patients was measured and estimated GFR (eGFR) was calculated using the equations [12] at baseline and 2 years after ultrasonography examination. As it is recommended by an international and Japanese working groups to use 30% to 40% decline in eGFR over 2 to 3 years as a surrogate for ESKD [13,14], we evaluated the percentage of decline in eGFR over 2 years and defined a 30% decline or initiation of maintenance dialysis during follow up as an endpoint in this study. This study was conducted in accordance with Declaration of Helsinki and approved by our institutional review board (approval number: 18A131).

### 2.2. Ultrasonography

All kidney ultrasonography was performed using standard grayscale B-mode imaging on an Aplio 500 Ultrasound System (Toshiba Medical Systems, Tochigi, Japan) with a 3.5-MHz convex transducer by one experienced physician (T.T., 8 years of clinical experience and ultrasound examination). Morphological measurements were performed as previously reported [15]. Briefly, kidney length, cortical thickness and medullary thickness were measured from the longitudinal images. Medullary thickness was the distance from the sinus fat to the corticomedullary junction, and cortical thickness was the distance from corticomedullary junction to the renal capsule (Figure 2). Cortical and medullary thicknesses were obtained from at least three different points. Parenchymal thickness was the sum of cortical and medullary thickness. Kidney thickness and width were also measured from the transverse images to calculate kidney volume [16]. All of the parameters were measured for the bilateral kidneys and the mean values of the left and right kidneys were used for the analysis. 

### 2.3. Statistical Analysis

Continuous variables were expressed as mean (SD) or median (range). Morphological parameters including kidney length, volume, cortical thickness and medullary thickness were investigated after adjustment for height. The Shapiro-Wilk test was used to assess normal distribution. The correlations of morphological parameters to renal function were analyzed using Pearson’s correlation coefficient and Spearman’s rank correlation coefficient. Multiple linear regression analysis, in which morphological parameters were selected with stepwise forward selection method in addition to baseline eGFR, was performed to investigate independent predictors of decline in renal function. Receiver operating characteristic curves were used to assess the predictive ability of morphological parameters for kidney dysfunction. Statistical analyses were performed using SPSS version 25 for Windows (IBM Japan, Tokyo, Japan) and StatFlex ver 6.0 for Windows (Artec, Osaka, Japan). A two-tailed *p*-value of less than 0.05 was considered statistically significant. 

## 3. Results

### 3.1. Patient’s Characteristics and Baseline Evaluation

The characteristics of the subjects are summarized in Table 1. Morphological parameters including length, cortical thickness and medullary thickness could be measured from all the patients, except that volumetric calculations were impossible in six patients. In order to eliminate influence from sex and body size, kidney morphological parameters were adjusted by height and body surface area as previously reported [15]. Referencing kidney morphological parameters to height diminished the sex differences, thus height-adjusted values were used thereafter in this study (Table 2). Univariate analysis at baseline revealed correlations of kidney length, volume and cortical thickness to eGFR (Table 3).

### 3.2. Association between Decline in Kidney Function and Kidney Morphology

Among 87 patients enrolled, one patient started maintenance hemodialysis during 2 years. Kidney functions of remaining 86 patients were evaluated and the association between baseline morphological parameters and decline in kidney function was analyzed. Univariate analysis showed that cortical thickness was correlated to the percent decline in eGFR over 2 years (*r* = 0.426, *p* < 0.001) (Figure 3). Multiple linear regression analysis, which included baseline eGFR and morphological parameters revealed cortical thickness as a strongest predictor for the percent decline in eGFR over 2 years (stβ = 0.458, *p* < 0.001) (Table 4).

We further investigated the predictive power of morphological assessments for renal dysfunction. Initiation of dialysis and the percent decline in eGFR over 2 years more than 30% were defined as an endpoint. Receiver operating characteristic analysis revealed that cortical thickness was the most accurate predictor than the other parameters with the area under the curve of 0.786, and the height-adjusted cortical thickness of 4.0 mm/cm was a cut off value with a sensitivity of 72.5% and a specificity of 80.0% (Figure 4).

## 4. Discussion

In the present study, we demonstrated that cortical thickness was correlated to decline in renal function. In addition, cortical thickness had the strongest predictive ability for renal dysfunction. This is the first longitudinal study investigating the predictive ability of morphological assessment by ultrasonography for renal dysfunction. 

Diagnosis and staging of kidney disease is based on altered renal function, which is defined by GFR. Because of the compensation by residual glomeruli, decline in renal function is hard to detect at the initial stage of kidney disease [17], moreover the compensatory response of the residual glomeruli increase intraglomerular pressure leading to further kidney injury [18,19]. The rate of decline in renal function depends on the initial eGFR; decreased eGFR itself accelerates the progression of kidney injury [20]. Therefore, latent parenchymal damage, which can be a risk for renal dysfunction, may exist in the absence of decline in GFR. New approaches based on the kidney structure have been suggested to improve the management of renal disease [21].

Recent studies have demonstrated the usefulness of the morphological assessment of kidney for diagnosing and managing CKD. Kidney length is a simple parameter, which can be easily measured by ultrasonography, and its decrease is frequently observed in advanced CKD [22,23]. Since kidney is composed of cortex, medulla, and non-parenchymal tissues such as sinus fat and calyces, detailed assessment of kidney size has been reported to be more useful than measuring kidney length. Meland et al. showed that cortical thickness measured by ultrasonography was positively correlated to renal function stronger than kidney length [6]. Another study using magnetic resonance imaging also showed a positive correlation between cortical thickness and renal function [24]. 

Despite that kidney structure is associated with renal function, little attention has been paid to the potential utility of morphological assessment for detecting earlier stages of CKD until recently. Small number of cross-sectional studies with enhanced computed tomography [7] or ultrasonography [15] suggested thinning of cortical thickness precedes that of medulla or length. 

In the present study, we performed a longitudinal assessment to clarify the usefulness of morphological assessment to predict renal dysfunction. The endpoint of this study was defined as more than 30% decline in eGFR over 2 years, which was revealed to be a surrogate for renal impairment [13,14]. Cortical thickness was well correlated to decline in renal function and showed area under the curve (AUC) >0.7, which indicated moderate accuracy for predict renal impairment. Together with these findings, height-adjusted cortical thickness is thought to be the best parameter for morphological assessment. Baseline eGFR did not correlate to the decline in renal function over 2 years. It has been shown, from a 10-year follow up study in a large population, that the slope of decline in renal function was larger in subjects with eGFR less than 40 mL/min/1.73 m^2^ compared to subjects with maintained eGFR [20]. However, among subjects with maintained renal function, the difference in the slope of decline was relatively small. Considering the patient’s characteristics enrolled in the present study and the small number of the patients investigated, baseline renal function might not be reflected to decline in renal function over short follow up period. These results suggest that cortical thickness possibly detected latent parenchymal damage.

There are some limitations in this study. GFR of the patient was not directly measured, but only estimated by serum creatinine. Patients with chronic glomerulonephritis were included in this study. However, active glomerulonephritis patients or nephrotic syndrome were excluded, and only stable patients were included in the study.

## 5. Conclusions

We provide new insights into the usefulness of measuring cortical thickness by ultrasonography to predict future renal impairment. This study could be of help in managing CKD.

## Figures and Tables

**Figure 1 jcm-07-00527-f001:**
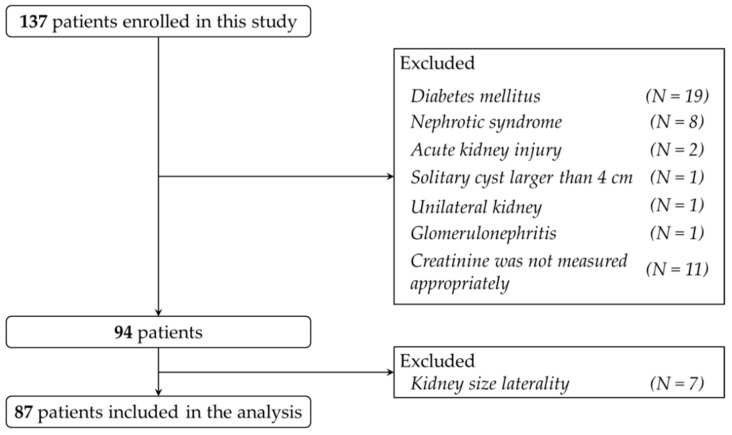
Study design. Of the 137 patients who underwent abdominal ultrasonography, 87 were included in the analysis.

**Figure 2 jcm-07-00527-f002:**
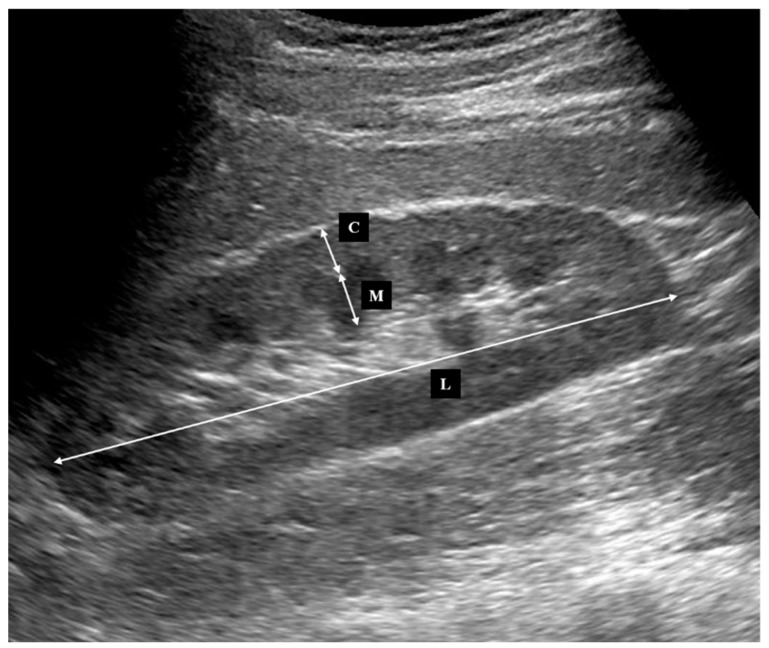
Longitudinal image of ultrasonography for the measurement. C, cortical thickness; M, medullary thickness; L, length.

**Figure 3 jcm-07-00527-f003:**
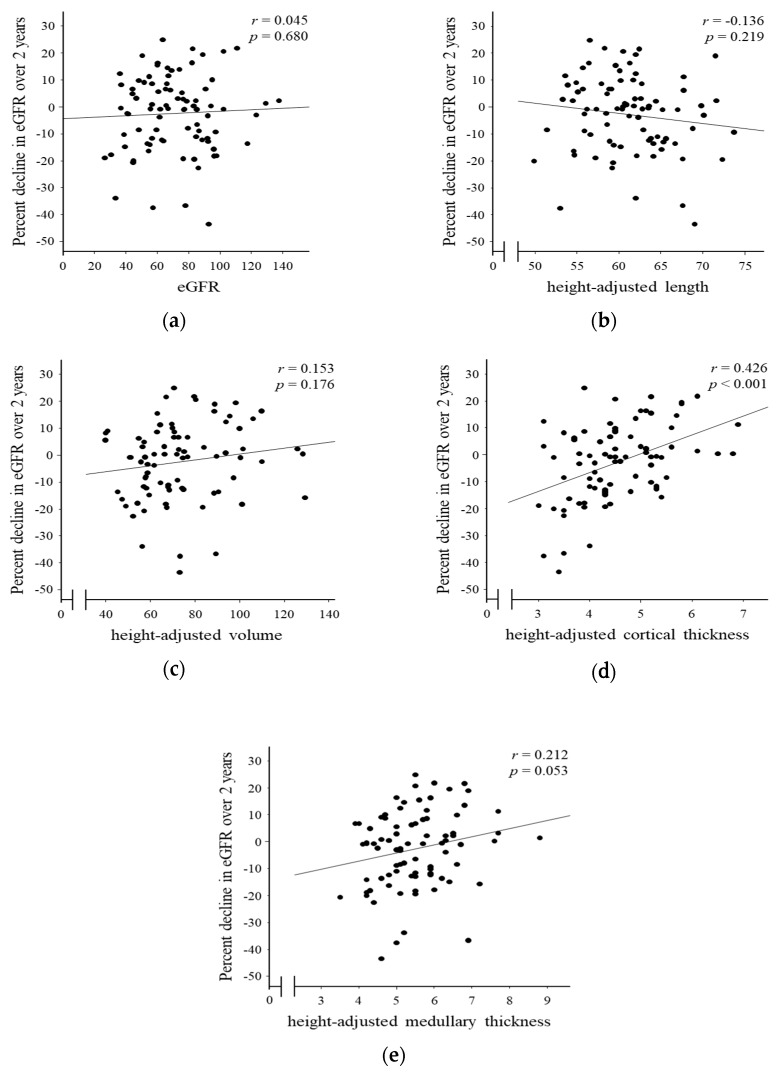
Correlation of morphological parameters by ultrasonography and decline in renal function over 2 years. Scatter plots comparing the decline in eGFR and baseline eGFR (**a**), kidney length (**b**), volume (**c**), cortical thickness (**d**) and medullary thickness (**e**). eGFR, estimated glomerular filtration rate.

**Figure 4 jcm-07-00527-f004:**
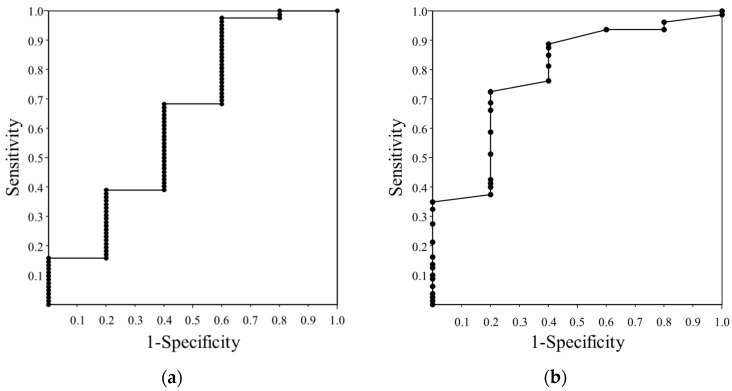
The ROC curve of baseline eGFR (**a**) and height-adjusted cortical thickness (**b**) for predicting renal dysfunction. The height-adjusted cortical thickness of 4.0 mm/cm was determined to be a diagnostic cut off value with a sensitivity of 72.5% and a specificity of 80.0%. ROC, receiver operating characteristic; eGFR, estimated glomerular filtration rate.

**Table 1 jcm-07-00527-t001:** Patient's characteristics at baseline.

Characteristic	Value
Number	87
Age, years	67.0 (28–95)
Male	44 (50.6)
Height, cm	159.0 (140–182)
Body weight, kg	55.7 (36.8–92.0)
Body surface area, m^2^	1.43 (1.21–2.12)
Serum creatinine, mg/dL	0.76 (0.35–4.72)
Estimated GFR, mL/min/1.73 m^2^	67.0 (10.2–137.7)
Kidney parameters	
Length, mm/cm	60.9 (49.9–73.7)
Volume, ×1000 mm^3^/cm	69.7 (39.6–129.3)
Cortical thickness, mm/cm	4.4 (3.0–6.9)
Medullary thickness, mm/cm	5.4 (3.5–8.8)

Kidney parameters were indexed to height, and average of the two kidneys is shown. Data are presented as the median (range), or *n* (%). GFR, glomerular filtration rate.

**Table 2 jcm-07-00527-t002:** Kidney morphological parameters with reference to body size.

Parameters	Reference	Average Values		
		Male	Female	M/F Ratio
Length, mm	None	100.4	94.4	1.063
	Height, cm	60.1	62.1	0.968
	BSA, m^2^	58.2	62.1	0.937
Volume, ×1000 mm^3^	None	138.0	97.6	1.414
	Height, cm	82.7	63.9	1.295
	BSA, m^2^	79.5	67.1	1.185
Cortical thickness, mm	None	7.7	6.9	1.123
	Height, cm	4.6	4.5	1.028
	BSA, m^2^	4.4	4.8	0.936
Medullary thickness, mm	None	9.1	8.4	1.072
	Height, cm	5.4	5.5	0.982
	BSA, m^2^	5.3	5.8	0.899

BSA, body surface area.

**Table 3 jcm-07-00527-t003:** Association between baseline eGFR and morphological parameters.

Variables	*r*	*p* Value
Length	0.439	<0.001
Volume	0.236	0.034
Cortical thickness	0.253	0.019
Medullary thickness	0.166	0.128

eGFR, estimated glomerular filtration rate.

**Table 4 jcm-07-00527-t004:** Multiple linear regression analysis: association between decline in kidney function and morphological parameters.

Variable	Std β	*p* Value
Cortical thickness	0.458	<0.001
Length	−0.208	0.058
eGFR	0.001	0.990

eGFR, estimated glomerular filtration rate.

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
