# Peer review of "Predictive Value of Cortical Thickness Measured by Ultrasonography for Renal Impairment: A Longitudinal Study in Chronic Kidney Disease"

_jcm, 2018, doi:10.3390/jcm7120527_

Reviewer 1 Report

Hoi et.al., were evaluated the morphological size of kidney by ultrasonography to find out whether the kidney size is a risk factor for the progression of chronic kidney disease (CKD). Authors are measured kidney length, cortical thickness and medullary thickness using the longitudinal images of ultrasonography from 109 patients. Data is analyzed thoroughly and the design of the study (Ex: Patients with unilateral kidney, tumor, PKD, acute kidney injury etc. were excluded from this study) is appropriate enough to achieve the aim mentioned in manuscript. Current article has interesting observation and beneficial to CKD patients, clinicians and researchers in kidney field. The overall study is suggesting the usefulness of measuring cortical thickness by ultrasonography to predict future renal impairment.

I had no specific suggestions for the authors with exception of the following:

1)      Line 42-43, Authors are suggested to re-frame the sentence to:  Kidney parenchyma is of composed of an outer cortex and inner medulla. Glomeruli, that define GFR, are distributed in the cortex. Therfore….

2)      Figure 2: indicate the units of X axis (Ex: Length (mm)). X and Y axis font size should be increased.

3)      Study is included data only from patients. Authors should have included data from few controls (normal subjects) to make the study more reliable

Author Response

Thank you very much for reviewing the manuscript and the constructive comments. We have carefully considered each of the comments and revised the manuscript. Point-by-point response to the comments are below;

1)      Line 42-43, Authors are suggested to re-frame the sentence to:  Kidney parenchyma is of composed of an outer cortex and inner medulla. Glomeruli, that define GFR, are distributed in the cortex. Therfore….

Sorry for the ambiguous expression. It would be more precise to separate outer / inner medulla, or further separate outer stripe / inner stripe of the outer medulla. However the authors would prefer to avoid complicated definition and refer the same wording as previous studies. Some points were changed according to the reviewer’s suggestion.

We have changed to “Kidney parenchyma is composed of cortex and medulla [5]. Glomeruli, that define GFR are….”

2)      Figure 2: indicate the units of X axis (Ex: Length (mm)). X and Y axis font size should be increased.

 Units and labels of the figures were changed to view easily.

3)      Study is included data only from patients. Authors should have included data from few controls (normal subjects) to make the study more reliable

 It is quite hard to add another subject to the population since this is a longitudinal study, however this study is still worth in the point of the longitudinal study design.

In addition to the changes mentioned above, some redundant sentences or references were changed.

Reviewer 2 Report

Study population and ultrasonography.

did the authors exclude patients with relevant differences between the two kidney? or with unilateral kidney disease?

Probably diabetic patients should be excluded ( or examined separately)

The diagnoses of kidney diseases should be reported.

The authors should report a flow diagram (STARD initiative) indicating the number of patients considered, those excluded and the final number considered for the study. 

In how many cases it was not possible to have the compete kidney measurements?

Table 1.

the authors should indicate in the caption that kidney parameters are indexed to height, and that the average of the two kidneys is reported.

to evaluate the additive value of kidney measurements, the authors should analyze also the correlation between baseline real function (Screat or eGFR) and the decline of renal function.

Therefore, Screat or eGFR should be introduced in multiple linear regression.

A ROC plot for baseline eGFR to predict renal function decline should be reported.

The discussion should modified accordingly to the results of these analyses.

As limitation of the study, the authors should report that GFR was not directly measured, but only estimated.

Author Response

Thank you very much for reviewing the manuscript and the constructive comments. We have carefully considered each of the comments and revised the manuscript accordingly.

Point-by-point response to the reviewer's comments are below;

did the authors exclude patients with relevant differences between the two kidney? or with unilateral kidney disease?

Patients with unilateral kidney were excluded from this study. We further evaluated the differences of the two kidneys and eliminated patients with relevant laterality. In detail, patients with size laterality of more than 2 cm were excluded according to a previous report. These changes were incorporated in the “Materials and Methods” section.

Probably diabetic patients should be excluded ( or examined separately)

We totally agree this point, since diabetes might have affected the morphological parameters. Because of the small number of the subject, we could not separately analyze patients with diabetes. Thus, according to the reviewer’s suggestion, we excluded patients with diabetes and performed the analyses. Patients population and results were modified according to the newly obtained results. A sentence mentioning the limitation of the study was removed according to this change.

The diagnoses of kidney diseases should be reported.

We modified and added some sentences together with the changes mentioned above to clearly define the criteria for subject recruitment in the “Materials and Methods” section. Also, study population was summarized in a newly added Figure.

The authors should report a flow diagram (STARD initiative) indicating the number of patients considered, those excluded and the final number considered for the study.

New figure representing the study population was added. The number of patients included and excluded was described.

In how many cases it was not possible to have the compete kidney measurements?

Kidney length, cortical thickness and medullary thickness could be measured from all the patients. Volume could not be calculated in 6 patients. This is incorporated in the “Results” section.

Table 1. the authors should indicate in the caption that kidney parameters are indexed to height, and that the average of the two kidneys is reported.

We added the sentence in the caption of Table 1.

to evaluate the additive value of kidney measurements, the authors should analyze also the correlation between baseline real function (Screat or eGFR) and the decline of renal function.

We agree to this point. The correlation between baseline eGFR and the decline of renal function was analyzed and shown in Figure 3 and Table 3.

Therefore, Screat or eGFR should be introduced in multiple linear regression.

According to the reviewer’s comments, we introduced baseline eGFR as explanation variable in the multiple linear regression analysis to evaluate the additive value of kidney measurements.

A ROC plot for baseline eGFR to predict renal function decline should be reported.

A ROC for baseline eGFR and decline in renal function was added in Figure 4.

The discussion should modified accordingly to the results of these analyses.

The main results obtained from additional analysis mentioned above was that cortical thickness was still the strongest predictor for decline in renal function, and baseline eGFR was not. We carefully interpreted the results and discussed this point. Modification is reflected in the “Discussion” section.

As limitation of the study, the authors should report that GFR was not directly measured, but only estimated.

We modified the limitation of this study mentioning this point.

In addition to the changes mentioned above, some redundant sentences or references were changed.

Round  2

Reviewer 2 Report

The authors modified the text according to the suggestions of the reviewer